# Glutathione and Glutaredoxin—Key Players in Cellular Redox Homeostasis and Signaling

**DOI:** 10.3390/antiox12081553

**Published:** 2023-08-03

**Authors:** Yuh-Cherng Chai, John J. Mieyal

**Affiliations:** 1Department of Chemistry, John Carroll University, University Heights, OH 44118, USA; ychai@jcu.edu; 2Department of Pharmacology, Case Western Reserve University, Cleveland, OH 44106, USA

**Keywords:** glutathione, glutaredoxin, glutathionylation, redox homeostasis, redox signaling, oxidative stress

## Abstract

This Special Issue of *Antioxidants* on Glutathione (GSH) and Glutaredoxin (Grx) was designed to collect review articles and original research studies focused on advancing the current understanding of the roles of the GSH/Grx system in cellular homeostasis and disease processes. The tripeptide glutathione (GSH) is the most abundant non-enzymatic antioxidant/nucleophilic molecule in cells. In addition to various metabolic reactions involving GSH and its oxidized counterpart GSSG, oxidative post-translational modification (PTM) of proteins has been a focal point of keen interest in the redox field over the last few decades. In particular, the S-glutathionylation of proteins (protein-SSG formation), i.e., mixed disulfides between GSH and protein thiols, has been studied extensively. This reversible PTM can act as a regulatory switch to interconvert inactive and active forms of proteins, thereby mediating cell signaling and redox homeostasis. The unique architecture of the GSH molecule enhances its relative abundance in cells and contributes to the glutathionyl specificity of the primary catalytic activity of the glutaredoxin enzymes, which play central roles in redox homeostasis and signaling, and in iron metabolism in eukaryotes and prokaryotes under physiological and pathophysiological conditions. The class-1 glutaredoxins are characterized as cytosolic GSH-dependent oxidoreductases that catalyze reversible protein S-glutathionylation specifically, thereby contributing to the regulation of redox signal transduction and/or the protection of protein thiols from irreversible oxidation. This Special Issue includes nine other articles: three original studies and six review papers. Together, these ten articles support the central theme that GSH/Grx is a unique system for regulating thiol-redox hemostasis and redox-signal transduction, and the dysregulation of the GSH/Grx system is implicated in the onset and progression of various diseases involving oxidative stress. Within this context, it is important to appreciate the complementary functions of the GSH/Grx and thioredoxin systems not only in thiol-disulfide regulation but also in reversible S-nitrosylation. Several potential clinical applications have emerged from a thorough understanding of the GSH/Grx redox regulatory system at the molecular level, and in various cell types in vitro and in vivo, including, among others, the concept that elevating Grx content/activity could serve as an anti-fibrotic intervention; and discovering small molecules that mimic the inhibitory effects of S-glutathionylation on dimer association could identify novel anti-viral agents that impact the key protease activities of the HIV and SARS-CoV-2 viruses. Thus, this Special Issue on Glutathione and Glutaredoxin has focused attention and advanced understanding of an important aspect of redox biology, as well as spawning questions worthy of future study.

## 1. Overview

Glutathione (GSH) is a tripeptide composed of three amino acids, glycine, cysteine, and glutamate. It exists ubiquitously in abundance within the cells of a broad spectrum of species and plays key functional roles. The nucleophilic cysteine-thiol group is the reactive principle of the molecule, mediating its various biological activities. Among small-peptide molecules, the structure of GSH is unique because the N-terminal glutamic acid and cysteine residues are linked via an unusual peptide bond involving the γ-carboxyl group of the glutamyl moiety rather than the α-carboxyl group, as in conventional peptide bonds [1] (Figure 1). Consequently, glutathione is the most abundant non-protein cellular thiol because this unique γ-bond renders glutathione relatively stable and resistant to intracellular degradation by proteases. Moreover, the unique architecture of the GSH molecule contributes to the glutathionyl specificity of the primary catalytic activity of glutaredoxin enzymes (see below).

Glutathione is present at concentrations ranging from 1 mM to 10 mM in the cytosolic compartment of most cells [2], but it also resides at lower concentrations in the subcellular organelles (endoplasmic reticulum, mitochondria, and nucleus). The molecule in the cytosol primarily exists in its reduced form (GSH), with a minor amount in the oxidized form (glutathione disulfide, GSSG); diminution in the GSH/GSSG ratio serves as a biomarker of oxidative stress. Intracellular variations in the GSH/GSSG ratio also correspond to distinct redox environments in the different cellular compartments, influencing their functions. For example, the actively reducing nature of the cytoplasm results in a GSH/GSSG ratio approaching or exceeding 100/1, so the formation of intra- and intermolecular protein disulfides is difficult and rare. In contrast, the GSH/GSSG ratio approaches 1/1 in the endoplasmic reticulum, providing a more oxidizing environment conducive to intramolecular disulfide formation, facilitating protein folding [3]. Accordingly, homeostatic enzymes and redox co-factors maintain particular GSH/GSSG ratios appropriate to sustain normal physiological processes.

GSH serves many critical homeostatic and regulatory functions in cells, including maintenance of a balanced redox environment (redox potential), defense against antioxidative stress, nucleophilic scavenging of reactive electrophiles, mediation of redox signaling, and regulation of cell growth versus cell death [4]. Two recent focused collections of articles have featured the special properties of the glutathione molecule and its potential therapeutic utility. Thus, a special issue of the journal *Molecules* was entitled “Glutathione: Chemistry and Biochemistry” [5], and a previous special issue of *Antioxidants* was titled “Glutathione in Health and Disease” [6]. In the current special issue of *Antioxidants* entitled “Glutaredoxin and Glutathione”, a collection of primary research articles and reviews are united by a central theme that pivots around the glutathionyl specificity of the thioltransferase catalytic activity of the glutaredoxin enzymes and the significant functions of glutathione in redox homeostasis and redox signaling, involving glutathionyl mixed-disulfide bonds with reactive cysteine residues on proteins (protein-S-glutathionylation).

## 2. Glutaredoxin Enzymes

The glutaredoxin oxidoreductase enzymes play central roles in redox homeostasis and signaling, and iron metabolism in eukaryotes and prokaryotes, under physiological and pathophysiological conditions [7,8,9,10,11,12,13,14]. The glutaredoxins are members of the thioredoxin superfamily and are structurally quite similar to thioredoxins and the N-terminal domain of glutathione transferases [9]; however, glutaredoxins have distinct catalytic properties. Most glutaredoxins belong to two major classes. Class I (Grx) includes the glutathione-dependent thiol:disulfide oxidoreductases [7,9,14,15,16], and class II includes the Grx-like proteins (Glp), which are inactive in standard oxidoreductase assays but serve as iron sensors, playing a critical role in glutathione-dependent delivery of iron–sulfur (Fe-S) clusters [8,10,11,12,17]. There are other minor Grx subfamilies, including several plant-specific Grx isoforms [18], which exert additional functions, such as the transcriptional regulation of petal development in flowers [19]. The current compendium of articles is focused on the role of class I glutaredoxins, in particular, and their roles in cellular regulation, in concert with glutathione.

## 3. Protein S-Glutathionylation

Although historically viewed as harmful byproducts of metabolism that are scavenged by antioxidants, reactive oxygen species (ROS) are also known to act as intracellular second messengers in redox-signaling pathways [20,21]. ROS serve this redox regulation function by modifying specific signaling proteins. In particular, ROS-mediated reversible oxidative modifications of the thiol moieties of protein-cysteine residues are characterized by their ability to modulate the protein activities, thereby propagating signal transduction and biological responses [22,23]. Such post-translational modifications of reactive protein-thiols include sulfenic acid formation (protein-SOH), nitrosylation (protein-SNO), S-glutathionylation (protein-SSG), and others. Considering the relative abundance of GSH in cells and the relative reactivity of various modified cysteine intermediates, it is likely that protein-SSG may represent the preponderant form of protein–cysteine modification, as depicted in the Figure 2 [24].

Although cysteine is one of the least abundant amino acids, it stands out as functionally distinct [25]. Even though it has a rather low occurrence in proteins in general, cysteine is often found in the functional sites of proteins, where it plays key roles in catalysis, regulation, secondary structure, etc. [26]. Moreover, cysteine residues often constitute metal-binding sites on proteins, facilitating the action of metal ions as cofactors [27]. These characteristic properties of cysteine residues are dependent on the physical nature and chemistry of the sulfhydryl moiety. Thus, the nucleophilicity of the thiol group is responsible for cysteine’s role in catalysis by enzymes like kinases and phosphatases. The redox reactivity of the thiol group enables cysteine to participate in structural thiol-disulfide interchange reactions affecting protein stability, and in oxidative posttranslational modifications that regulate function and propagate signaling pathways [28]. Indeed, reversible S-glutathionylation of protein-SH groups in cells can serve as a protective mechanism for proteins against irreversible oxidative modification (e.g., sulfonation) or serve a regulatory role as an intermediate in signal transduction. The formation of protein-SSG mixed disulfides introduces additional local ionic charges on the protein, analogous to the effect of protein phosphorylation, potentially altering the function of the protein in a reversible fashion (see Section 4, below). This reversible feature of S-glutathionylation accounts for its ability to act as a “redox switch,” which regulates the activity and/or translocation of a protein transiently [29,30].

About 214,000 cysteines are encoded in the human genome [31]. Proteins may have exposed cysteine residues on the surface within the aqueous environment [32], or may be embedded deep within the more hydrophobic globular domains. In an aqueous milieu, the thiol group of cysteine is prone to deprotonation, in equilibrium with the negatively charged thiolate moiety. Both the protonated and unprotonated forms of cysteine have non-bonded pairs of electrons, consistent with nucleophilicity, but the thiolate anions are much more reactive. The ratio (thiol/thiolate) at any pH condition is related to the thiol pK_a_, which for a typical cysteine residue is near 8.5 [25]; accordingly, only a small fraction of cysteine residues would be negatively charged at physiological pH (7.4). However, the local microenvironment (e.g., neighboring cations) can strongly affect the pK_a_ values of protein thiols over a wide range [26]. Although a lower pK_a_ value corresponds to a higher thiolate amount at neutral pH, it is important to note that the relative rates of cysteine-mediated reactions also depend on other factors [33].

Among the mechanisms of formation of protein-SSG adducts, thiol-disulfide exchange is one of the most extensively studied [7]; however, it relies primarily on the redox state of cellular glutathione. According to this mechanism, the intracellular GSH/GSSG ratio dictates the extent of protein S-glutathionylation ([Protein-SSG]/[Protein-SH]), and the equilibrium constant for the reaction (K_mix_, Equation (1)) corresponds to the oxidation potential for the formation of the mixed disulfide (protein-SSG):

Protein-SH + GSSG ⇌ Protein-SSG + GSH,
(1)$$K_{mix}=\frac{\left[Protein-SSG\right][GSH]}{\left[Protein-SH\right][GSSG]}$$

The K_mix_ value for most cysteine residues is approximately 1.0, so the GSH/GSSG ratio would have to be decreased greatly in order to favor the formation of protein-SSG [7]. However, as described above, the cytosolic GSH/GSSG ratio usually remains very high, even under pronounced oxidative stress conditions [34], rendering the thiol-disulfide exchange mechanism with GSSG as the mediator thermodynamically unfavorable. Accordingly, the documentation of S-glutathionylation of particular proteins in the presence of artificially high concentrations of GSSG in vitro may not reflect what would actually occur in vivo. Instead, other mechanisms involving reactive thiol derivates, such as thiyl radicals and others (see Figure 1), are more likely than GSSG to mediate protein-SSG formation in vivo. Thiyl radicals are highly reactive species that are generated by hydrogen atom abstraction from the -SH moiety [35,36]. Several types of free radical species can oxidize thiols to thiyl radicals, including superoxide (O_2_^•–^), and hydroxyl (^•^OH), peroxyl, and phenoxyl radicals, among others [37,38]. The generation of glutathione-thiyl radicals or protein-thiyl radicals leads to protein-SSG formation via a radical recombination mechanism or through the reaction of a radical with a thiolate, followed by a reaction with oxygen [24]. In support of this potential mechanism of S-glutathionylation, several proteins were shown to be glutathionylated in vitro using glutathione-thiyl radical generating systems (horseradish peroxidase (HRP)/H_2_O_2_ + GSH, or Fe^2+^/ADP/H_2_O_2_ + GSH) [39]. Moreover, Kang et al. [40] reported that a burst of superoxide produced by mitochondria led to thiyl radical formation and S-glutathionylation of the Complex I protein, consistent with the concept that thiyl radicals may serve as reactive intermediates promoting S-glutathionylation in vivo. In addition, there is evidence that such thiyl radical-mediated protein-SSG formation can be catalyzed by glutaredoxin [24].

Another important mechanism that may mediate protein-SSG formation in vivo involves another type of reactive thiol derivate, namely sulfenic acid. Several different oxidants, including hydrogen peroxide, alkyl hydroperoxides, peroxynitrite, hypochlorous acid, and chloramines, are implicated in mediating the conversion of protein-thiolates to sulfenic acids (protein-SOH) [41], but increased exposure to these oxidants can lead to further oxidation and irreversible modification (sulfinic and sulfonic acids), or promote reactions with neighboring thiols to form disulfides [41]. Sulfenic acid formation was found to be a regulatory mechanism for many proteins [25]. However, many of the studies supporting this interpretation were performed in the absence of GSH. Thus, although protein-SOH formation may be the initial oxidative modification of cysteine residues, subsequent S-glutathionylation may occur in the presence of abundant GSH in cells, serving as a more stable redox intermediate in signal transduction [24]. This scenario was demonstrated, for example, in the case of BiP, a molecular chaperone. Originally thought to be regulated by sulfenic acid formation, BiP was later reported to undergo S-glutathionylation in a step-wise fashion involving a sulfenic acid intermediate [42]. An analogous scenario pertains to reactions involving reactive nitrogen species related to nitric oxide. Thus, some regulatory mechanisms originally attributed to nitric oxide signaling via protein-SNO were later understood to involve the sequential formation of protein-SSG. For example, the regulation of the SERCA calcium pump in cardiac cells was characterized as being mediated by NO-stimulated S-glutathionylation [43].

In addition to spontaneous non-enzymatic chemical reactions leading to protein-SSG formation in cells, several enzymes have been identified as potential catalysts for the S-glutathionylation of protein-SH groups, especially the Glutathione S-transferases (GSTs). Known as a family of detoxification enzymes, GSTs are characterized by their catalysis of the adduction of nucleophilic GSH to chemically reactive electrophiles [44]. In this context, the Pi class of GSTs (GSTP) in particular was identified as potentially contributing to the mechanisms of resistance to electrophilic carcinogens and chemotherapeutic drugs. Thus, elevated levels of GSTP were reported in solid tumors, but their actual function in catalyzing detoxification in this context remains uncertain [45]. The potential involvement of GSTP in S-glutathionylation was first recognized because GSTP was found to catalyze the adduction of GSH to peroxiredoxin IV, a lipid peroxidase enzyme that scavenges lipid hydroperoxides [46]. Subsequently, GSTP was implicated in playing a key role under oxidative-stress conditions in mediating the S-glutathionylation of several proteins, both in vitro and in vivo [47,48]. GSTP-knockout mice treated with PABA/NO (nitrosylating agent) were reported by Townsend et al. [44] to have decreased protein-SSG levels; however, when GSTP was present, the rate of S-glutathionylation was reported to be enhanced significantly. Consistent with these findings, GSTP knockdown in HEK293 cells was correlated with decreased protein-SSG levels [49]. Therefore, GSTP may promote protein-SSG formation with a wide variety of thiol-proteins, and the high expression of GSTP in cancer cells is likely associated with an elevated state of oxidative stress in these cells [45,50]. Indeed, in a broad-based proteomic study of mouse liver, utilizing tandem mass spectroscopy, McGarry et al. [51] reported extensive coverage of S-glutathionylated proteins, highlighting the heavy involvement of mitochondrial and Krebs-cycle enzymes; moreover, using GSTP knockout mice, they documented the potential for enzymatic mediation of protein-SSG formation.

As described above, the normal cytosolic milieu is characterized by high levels of GSH and a high GSH/GSSG ratio; accordingly, Grx1 primarily catalyzes deglutathionylation of protein-SSG substrates under these conditions (removal of glutathione from protein-SSG; see next subsection for more detail). During elevated oxidative stress, however, glutathione may be converted to oxidized forms, GSSG, GS-OH, GS-NO, and GS^•^, etc.; under such conditions, Grx1 is found to catalyze the backward reaction, namely the formation of protein-SSG [7,52]. Mieyal et al. recognized that the uniquely low pK_a_ of Cys22 at the active site of Grx1 [7] would serve to stabilize the disulfide-anion radical form of the enzyme (i.e., Grx1-SSG^•−^) in the presence of glutathionyl radicals (GS^•^); accordingly, they hypothesized that Grx1 could catalyze protein-SSG formation under such conditions via the enzyme-radical intermediate. Indeed, using the glutathionyl radical as the proximal glutathionyl donor, the authors of [39] observed that Grx1 promoted the formation of GAPDH-SSG, actin-SSG, and PTP1B-SSG. Moreover, O_2_ was found to be a competitive inhibitor of the S-glutathionylation reactions by intercepting the radical from Grx1-SSG^•−^ to form O_2_^•–^; this finding reinforces the concept that the redox environment likely determines the mode of catalysis by Grx1 [39].

## 4. Deglutathionylation of Protein-SSG

As mentioned above, S-glutathionylation of proteins is dynamic and reversible, so the steady-state level of protein-SSG under various conditions depends on the relative rates of glutathionylation (formation) and deglutathionylation (breakdown) (Figure 3), providing another level of regulation for cellular processes [53]. Specific binding sites for glutathione have been characterized on glutaredoxin, which is understood to be the primary catalyst of deglutathionylation [54]. As noted, glutaredoxin belongs to the thioredoxin superfamily of enzymes, which includes thioredoxins, glutathione peroxidases, glutathione S-transferases, and protein disulfide isomerases (PDI) [55]. Structurally, all of these proteins display a thioredoxin-fold motif, which is essential for their redox function; however, they otherwise possess only low sequence similarity [54].

Glutaredoxin isoforms exist in various subcellular compartments in eukaryotes and also in prokaryotes. As described under Section 1, mammalian glutaredoxins are broadly classified into two subfamilies based on their active site sequences and relative activities/functions, with Grx1 and Grx2 representing the major forms of each family [56]. Human Grx1 is localized in the cytosol and mitochondrial intermembrane space, whereas Grx2 is primarily localized in mitochondria. Grx1 is better characterized and reported to catalyze most of the deglutathionylating activity in mammalian cells [57,58]. Although Grx1 is not an essential protein since knockout mice are viable with a life span similar to wild-type mice [54], it is implicated broadly in cellular functions and defense against disease. For example, the level of protein S-glutathionylation is linked to the development of diseases in Grx1-knockout models [54], and the high concentrations of protein-SSG could be reversed by the exogenous administration of recombinant Grx1. Many previous reviews have considered the role of Grx1 in health and disease [59,60,61,62,63]. Human Grx2 is only 34% identical to Grx1 [64] and is about 20 times less abundant than Grx1 [65]. Mice lacking Grx2 are viable; however, they develop heart hypertrophy and fibrosis and become hypersensitive [66]. Although monomeric Grx2 can catalyze the reduction of S-glutathionylated proteins like Grx1, the catalytic efficiency of Grx1 is greater [24], and these two proteins behave differently in response to oxidative stress. Grx1 can be deactivated when cysteine residues are oxidatively modified, whereas Grx2 can be activated [67]. Grx2 is relatively insensitive to oxidative inactivation compared to other thiol-redox proteins because Grx2 can form Fe-S clusters [68]. Oxidative stress causes the degradation of the clusters and formation of active monomeric Grx2; therefore, Fe-S clusters function as a sensor for Grx2.

Grx1 and Grx2 have both been shown to selectivity catalyze dethiolation of S-glutathionylated substrates with GSH as co-substrate. The glutaredoxin substrate specificity was determined by a carefully designed experiment using various protein mixed-disulfide substrates with glutathione- and non-glutathione-containing thiols [69]. Grx effectively catalyzed only the glutathione-containing substrates but was ineffective for other substrates. These two isozymes share an analogous catalytic mechanism for deglutathionylation involving a nucleophilic, double-displacement (ping-pong) sequence, wherein only the N-terminal cysteine residue of the active-site CPYC motif participates in catalysis (monothiol mechanism) [70]. The main distinction between them is the decreased catalytic efficiency (k_cat_/K_M_) of Grx2, primarily due to a decreased k_cat_ [70]. The lower Kcat of Grx2 is because of the catalytic cysteine’s higher pKa and a decreased nucleophilicity enhancement of the second substrate, GSH [70]. Like Grx1, Grx2 exhibits GS-thiyl radical (GS^•^) scavenging activity, promoting the S-glutathionylation of various proteins [70].

Most glutaredoxin isoforms have analogous active-site motifs, displaying variations in the general 4-amino acid sequence Cys-X-X-Cys, which is redox reactive [63]. The primary activity of glutaredoxin involves nucleophilic displacement reactions corresponding to thiol-disulfide exchange. Monothiol and dithiol mechanisms have been described for different substrates; however, the monothiol mechanism is generally considered to be the preponderant mechanism for deglutathionylation [7]. The dithiol mechanism is used to explain the supporting role of Grx in the turnover of ribonucleotide reductase and DNA synthesis. According to the monothiol mechanism, the protein-thiol sulfur atom of the protein-SSG is attacked by the active-site thiolate anion of glutaredoxin, forming the intermediate Grx-SSG and releasing protein-SH. The second step of this nucleophilic double-displacement mechanism involves an attack on the distal sulfur atom of the Grx-SSG enzyme intermediate by GSH to produce GSSG and release the free enzyme Grx-S^–^. Then, the GSSG product is subsequently reduced by NADPH and glutathione reductase [7]. In the dithiol mechanism, the first step is depicted as the formation of a protein-Grx mixed disulfide (protein-SS-Grx) involving the N-terminal cysteine and releasing GSH [71,72]. The second step is the intramolecular attack of the C-terminal cysteine on the mixed disulfide to displace the reduced protein-SH and form the intramolecular disulfide form of Grx. Then, two molecules of GSH complete the reaction in a step-wise manner; the first GSH displaces the intramolecular disulfide to form the Grx-SSG intermediate, and the second GSH leads to the formation of GSSG and the free Grx-(SH)_2_ enzyme. Evidentially, the monothiol mechanism is the more efficient catalytic cycle.

Human glutaredoxin 2 (hGrx2) was the first member of the glutaredoxin family identified as an iron–sulfur protein. The Fe-S cofactor was shown to be bridged between two monomers via the N-terminal active site cysteine residues and two non-covalently bound GSH molecules [68]. The bound GSH was in exchange with the free GSH pool and played an essential role in stabilizing the Fe-S cluster. The dimeric form of hGrx2 was enzymatically inactive [68]. Under oxidative stress, an altered GSH/GSSG ratio limiting reduced GSH availability to maintain the Fe-S coordination resulted in the degradation of the cluster and formation of the enzymatically active Grx2 monomer. These results were interpreted to suggest that the Fe-S cluster functions as a redox sensor for Grx2 activity under oxidative stress. As explained earlier, other members of the broad glutaredoxin family have been characterized as participating in iron–sulfur homeostasis [10,11,17].

## 5. Highlights of the Special Issue on Glutathione and Glutaredoxin

In the following paragraphs, we provide brief descriptions highlighting the key findings featured in each of the original research articles and reviews included in this compendium Special Issue of *Antioxidants* on Glutathione and Glutaredoxin.

Two of the articles [73,74] feature the specific roles of Grx1 in liver fibrosis and lung fibrosis. Importantly, the data presented in these papers suggest a potential therapeutic role for Grx1 as an anti-fibrotic agent. Thus, Reiko Matsui and her coworkers [73] showed that the overexpression of Grx1 inhibits age-induced hepatic apoptosis and liver fibrosis in mice. On the other hand, high-fat and high-fructose diet-induced non-alcoholic steatohepatitis (NASH) leads to the downregulation of Grx1 and higher levels of S-glutathionylated proteins in the liver; overexpression of Grx-1 significantly decreases the expression of Zbtb16 and leads to the reversal of NASH progression by attenuating inflammatory and fibrotic processes. Although the primary role of Zbtb16 in hepatocytes is unknown, the current study highlights it as an important redox-sensitive protein, whose expression is regulated by Grx1. Certainly, further study of Zbtb16 function is warranted.

Yvonne Janssen-Heininger and her coworkers [74] reported that Grx1 activity was directly correlated with lung function, whereas protein-SSG accumulation was inversely correlated with lung function in subjects with idiopathic pulmonary fibrosis. Epithelial cells lacking Grx1 were more susceptible to Fas-ligand-induced apoptosis and displayed elevated FAS-SSG compared to wild-type controls, whereas the overexpression of Grx1 attenuated epithelial cell apoptosis in association with diminished Fas-SSG. Several metabolites in the purine, creatine, and other metabolic pathways, including inosine monophosphate, spermidine, and others, were consistently released from multiple cell types subjected to various apoptotic stimuli, including Fas. These findings establish a link between Grx1 activity and the modulation of multiple pathways that regulate the synthesis and utilization of diverse metabolites released by apoptotic cells. Further study is necessary to unravel which specific protein-SSG targets besides Fas-SSG may be responsible for the mediation of lung epithelial cell apoptosis and whether upregulation of Grx1 could be translated for the therapy of pulmonary fibrosis.

The two articles described above [73,74] implicate a potential therapeutic role for the upregulation of Grx1 in the context of fibrotic diseases of the liver and lung. However, the role of Grx1 in different disease contexts may be different. For example, with regard to Parkinson’s disease (PD), a dilemma arises when considering Grx1 as a PD-therapeutic target, namely whether to stimulate its upregulation for neuroprotection or to inhibit its proinflammatory activity [75]. The concept of inhibiting Grx1 activity as a therapeutic approach has also emerged in the context of viral diseases, where the S-glutathionylation of specific proteins has been implicated in the regulation of viral infections. A review article included in this compendium [76] considers how S-glutathionylation modulates the activity of key protease enzymes responsible for viral replication.

David Davis, Robert Yarchoan, and their coworkers [76] reported that protein S-glutathionylation regulates retroviral protease activity, including human immunodeficiency virus type 1 (HIV-1), human T-cell leukemia virus (HTLV-1), and SARS-CoV-2 proteases. In general, particular proteases of each virus are required for viral maturation, and the protease activities are dependent on the dimeric forms of the enzymes, which can be altered by site-selective S-glutathionylation. For example, HIV-1 protease contains two cysteine residues, Cys67 and Cys95, with low pKa values. S-glutathionylation of Cys67 (C95A-mutant protease) increased the activity by two-fold. On the contrary, S-glutathionylation of Cys95 completely inhibits the activity by disrupting the dimerization of the protease. The oxidation of Cys95 in immature virions impaired viral maturation, and this effect can be reversed by disulfide reduction. Grx1 catalyzes the deglutathionylation of Cys95 and restores protease activity much more efficiently than it deglutathionylates Cys67. Likewise, HTLV-1 protease activity can be regulated by S-glutathionylation and activity restored by Grx1. The S-glutathionylation of Cys95 in HIV-1 protease and Cys109 in HLTV-1 protease sterically interfere with beta-sheet formation and dimerization, according to crystal structure studies. More recently, studies of the main protease (M^pro^) of SARS-CoV-2 show that it contains 12 Cys residues in each monomer; and, analogous to HIV-1 and HTLV-1, M^pro^ requires dimerization for enzyme activity. The S-glutathionylation of M^pro^-Cys300 inhibits dimerization and protease activity, reversible by Grx1. The M^pro^ of SARS-CoV-2 is encoded as part of two proteins, pp1a and pp1ab, and it remains unclear whether the inhibitory S-glutathionylation occurs in pp1a and/or pp1ab. Furthermore, it is conceivable that multiple types of reversible modifications of cysteine (glutathionylation, nitrosylation, or others) could occur under oxidative stress conditions in cells during viral infections. Hence, much is yet to be learned about the parallel or sequential relationship among the cysteine modifications of the viral proteases within the infected host cells, the extent to which these modifications affect the rate of viral replication, and how these modifications are regulated in situ. These basic studies suggest that the dimer interfaces of the viral proteases are prime targets for antiviral therapy and that simultaneous inhibition of the redox enzymes like Grx1, which reverse cysteine modifications, could serve as a synergistic adjunct therapy.

As described above, the brain is a major site where oxidative stress has been implicated in disease onset and progression, including various neurodegenerative disorders, such as Parkinson’s disease and Alzheimer’s disease. The brain is particularly susceptible to oxidative stress due to its high oxygen consumption, and glutathione turnover in the brain is not as efficient as in the liver. In the review contributed to this compendium by Vijiayalakshmi Ravindranath’s research group [77], the augmentation of the glutathione and glutaredoxin systems is considered an approach to protect the brain from oxidative stress damage during aging. Two significant characteristics of Parkinson’s disease (PD) are the loss of dopaminergic neurons and the accumulation of Lewy bodies (aggregates of a-synuclein). Dopaminergic neurons are rich in neuromelanin and iron, which can engage in Fenton chemistry and generate ROS. Mitochondrial dysfunction is implicated in PD, and Grx1 is essential for maintaining mitochondrial functions in the brain treated with MPTP(1-methyl-4-phenyl-1,2,3,6-tetrahydropyridine). MTTP, a mitochondrial-Complex I poison, is the most common neurotoxin that induces PD-like pathology by accelerating ROS production and GSH depletion. Moreover, a significant downregulation of Grx1 mRNA was observed in the substantia nigra region of human PD autopsy tissue. In addition to Grx1, the serine/threonine kinase Akt1 is also critical for dopaminergic cell survival. The oxidation of cysteine residues in Akt1 leads to dephosphorylation and inhibition of its activity in MPTP-treated mice. Also, the focal delivery of diamide, a well-known thiol oxidizing molecule that promotes protein-SSG formation, to the substantia nigra triggers Parkinsonism phenotypes; within this scenario, it is likely that Akt1-SSG formation leads to its inactivation.

In analogous studies of Alzheimer’s disease (AD), increased S-glutathionylation of proteins has been observed in brain samples from AD patients, and actin-SSG is one of the target proteins. Importantly, the regulation of the dynamic polymerization of actin is vital to the function of neural synapses, affecting memory and learning, and it is noteworthy that overexpression of Grx1 in primary cortical culture leads to the restoration of F-actin nano-assembly and spine morphology. Furthermore, previous cellular studies have established that Grx1 is essential for the deglutathionylation of actin-SSG and regulation of actin’s functional dynamics [78]. Thus, it is conceivable that enhancing the level of Grx1 in AD brains could maintain synaptic plasticity and forestall memory deficits. As most antioxidants do not cross the blood–brain barrier to reverse the effect of oxidative stress on the brain, common antioxidants, like α-lipoic acid and vitamins E and C, as well as precursors of GSH, have displayed only minimal beneficial effects, if any, upon administration to patients. Alternatively, augmentation of the intrinsic system, such as increasing the level of Grx1 early in disease progression, may be a more effective approach to combat oxidative stress.

Continuing the focus on mitochondria as an engine of oxidative stress, Ryan Mailloux and coworkers in an original research contribution to the Special Issue [79] reported S-glutathionylation of the NDUFS1 subunit of Complex 1 of the electron transport chain in liver mitochondria. This modification caused inhibition of Complex I activity and increased the generation of hydrogen peroxide, using glycerol-3-phosphate or proline as fuel during reverse electron transfer. Adding a reducing agent to the reaction system was found to reverse both the inhibition and the ROS accumulation. Interestingly, two other mitochondria proteins, glycerol-3-phosphate dehydrogenase and proline dehydrogenase, were not S-glutathionylated under conditions similar to those that led to NDUFS1-SSG formation, even though these two proteins also produce ROS. Complex III also undergoes S-glutathionylation with an effect similar to that of Complex I, but whether this effect is due to the Complex III modification or glutathionylation of complexes upstream from ubquinone–cytochrome C oxidoreductase remains to be elucidated.

Both Grx and protein disulfide isomerase (PDI) belong to the thioredoxin superfamily, contain CXXC sequences at their respective active sites, and utilize glutathione as a co-substrate. Although these two proteins have a similar 3D structure and thiol-disulfide catalytic mechanism, their cellular environments are quite different; thus, Grx catalyzes the reduction of glutathionyl mixed disulfides, whereas PDI catalyzes the formation of intramolecular disulfides. Ruddock and coworkers have been studying these two enzymes, and they contributed an original research article to this compendium [80] in which they reported the mutation of the PDI CXXC motif, converting histidine to tyrosine or phenylalanine, thereby changing the signature sequence to be more similar to that of class I glutaredoxins. These substitutions for histidine were found to change the binding affinity of PDI for its protein substrates and glutathione. Although the mutations did not significantly alter the overall structure of PDI, the mutant proteins displayed increased rates of GSH-dependent reduction or GSSG-dependent oxidation at the PDI active site. The His→Tyr and His→Phe mutants also showed an increased rate of deglutathionylation. This study documents that the specific CXXC sequences at the active sites of Grx and PDI are largely responsible for distinguishing their physiological functions within their quite different intracellular redox environments.

A very important aspect of understanding the impact of post-translational modification of protein–cysteine residues on the functional consequences is the validity and accuracy of the analytical methods used to identify the PTMs. In this compendium, Wei-Jun Qian and coworkers [81] have contributed a review article focused on the current biochemical and analytical approaches to characterize protein S-glutathionylation at both the proteome level and with individual proteins, including a perspective on studies of the functional impacts. Importantly, they highlight the challenges of some of the methods and consider ways to overcome these difficulties in future work. For example, it is difficult to directly verify the S-glutathionylation of specific cysteinyl moieties in their native environment in situ using mass spectroscopic (MS) approaches. The glutathionyl moiety on a Cys residue can undergo fragmentation in the MS/MS mode of analysis, leading to neutral losses via cleavage of the peptide bond. Another challenge is interpreting the S-glutathionylation data due to the complexity of the oxidative modification of cysteine. As mentioned above, a redox-sensitive cysteine residue on a protein can undergo multiple types of modifications (see Figure 2), so the ultimate questions are many, including which of these modifications is responsible for the functional changes, which modifications represent transient precursors versus actual intermediates in signal transduction pathways, and if S-glutathionylation might represent a more stable endpoint rather than a functional intermediate. Further development of techniques that quantify the stoichiometry and relative time courses of multiple forms of thiol modification would significantly improve our understanding of the details of redox regulation. The crosstalk between different types of thiol PTMs has long been overlooked due to the need for methods to characterize them. Nevertheless, the interplay between redox PTMs and other PTMs (e.g., phosphorylation and ubiquitination) is an exciting area to explore. Investigating the impact of the PTMs on protein structures and functions is currently a labor-intensive and slow process; however, advanced computational approaches could be employed to integrate the PTM sites and possibly elucidate their functional significance more rapidly. Structural studies and molecular dynamics analyses could facilitate the prioritization of functionally important S-glutathionylation sites and guide the design of validating experiments to distinguish among other types of cysteine-PTMs.

S-nitrosylation is a form of cysteine modification that has been broadly implicated in signal transduction, cellular homeostasis, and disease. Rajib Sengupta and coworkers have a special interest in S-nitrosylation as a regulatory mechanism. In their review that contributed to this Special Issue [82], they focused on the mechanisms of denitrosylation of protein-SNO and the enzymes that catalyze it, analogous to the catalysis of deglutathionylation by glutaredoxin. S-nitrosylated proteins can be denitrosylated by GSH within the range of the physiological concentrations of GSH (5–10 mM), except for a few proteins, such as caspase 3. It is conceivable that those proteins that are resistant to dentrosylation by GSH alone may be more likely to be regulated by reversible S-nitrosylation, requiring enzymatic reversal. GSH denitrosylates proteins in two ways: by displacement of the NO moiety, forming protein-SSG, or by transnitrosylation, forming GS-NO. A key element in this mechanism is GS-NO reductase (GSNOR), which is a significant regulator of the relative cellular levels of protein-SNO and GS-NO. Thus, GSNOR does not directly denitrosylate proteins; instead, it decreases protein-SNO levels by depleting GS-NO. Caspase3-SNO, which is not denitrosylated by GSH/GSNOR, is reduced by the thioredoxin (Trx) system. Although the primary catalytic function of thioredoxin in reducing intramolecular disulfide bonds in proteins is well understood, the relative role of Trx as a denitrosylation catalyst for protein-SNO has yet to be well characterized in mammalian cells. In addition to its deglutathionylation function, glutaredoxin can also denitrosylate proteins-SNO, and the mechanism of denitrosylation is expected to be analogous to the monothiol or dithiol mechanisms of deglutathionylation of protein-SSG by Grx. The authors cited studies in which knockdown, knockout, or inhibition of components of either the Trx or Grx systems led to an increased, reversible susceptibility to nitrosative/oxidative stress-induced cell damage. Thus, these two seemingly “redundant” redox systems appear to be essential for ensuring cell survival under stress. The take-home message of this review is the importance of the functional overlap of the glutaredoxin/glutathione and thioredoxin systems and their complementary functions as denitrosylation catalysts. Delineating the crosstalk among the redoxin enzymes is viewed as essential for understanding the extent of cell survival under oxidative/nitrosative stress in various diseases such as cardiovascular and neurodegenerative diseases.

The final article to be included in this compendium was contributed by Marjorie Lou [83], whose career has been devoted to characterizing redox homeostasis in the lens of the eye. In this review, she described the roles of glutathione and glutaredoxin in redox regulation and cell signaling, focusing on glutaredoxin in signal transduction in the lens and the role of ROS as growth factors. Documenting the role of glutaredoxin in growth factor-stimulated cell proliferation is a very important landmark in lens research, along with deciphering its role in lens aging and disease. Thus, glutaredoxin-knockout mice developed cataracts four months sooner than wild-type mice, and the lenses of KO mice showed corresponding increases in protein S-glutathionylation and decreased GSH levels. The glutaredoxin activity and GSH were significantly decreased in aged human lenses, and the loss of glutaredoxin activity and GSH was correlated with the severity of the lens opacity. Several areas of lens research warrant further consideration in addressing these key questions. For example, how important are glutaredoxin-2 and the thioredoxin/thioredoxin reductase system in redox regulation in the lens? What is the role of cysteine and protein-SS-cysteine mixed disulfide accumulation in relation to lens transparency? What is the role of redox regulation in cell differentiation, cell-to-cell communication, and lens development? The review by Lou et al. provides a strong foundation for future research.

## 6. Concluding Statement

The non-protein glutathione and the enzyme glutaredoxin are partners in performing critical cell functions. The data and information from the ten articles comprising this Special Issue support the central theme that GSH/Grx is a unique system for regulating thiol-redox homeostasis and redox-signal transduction; and dysregulation of the GSH/Grx system is implicated in the onset and progression of various diseases involving oxidative stress. Within this context, it is important to appreciate the complementary functions of the GSH/Grx and thioredoxin systems, not only in thiol-disulfide regulation but also in reversible S-nitrosylation. Several potential clinical applications have emerged from a thorough understanding of the GSH/Grx redox regulatory system at the molecular level, and in various cell types in vitro and in vivo, including, among others, the concept that elevating Grx content/activity could serve as an anti-fibrotic intervention, and discovering small molecules that mimic the inhibitory effects of S-glutathionylation on dimer association could identify novel anti-viral agents that impact the key protease activities of HIV and SARS-CoV-2 viruses. Thus, this Special Issue on Glutathione and Glutaredoxin has focused attention and advanced the understanding of an important aspect of redox biology, as well as spawning many exciting questions worthy of further study.

## Figures and Tables

**Figure 1 antioxidants-12-01553-f001:**
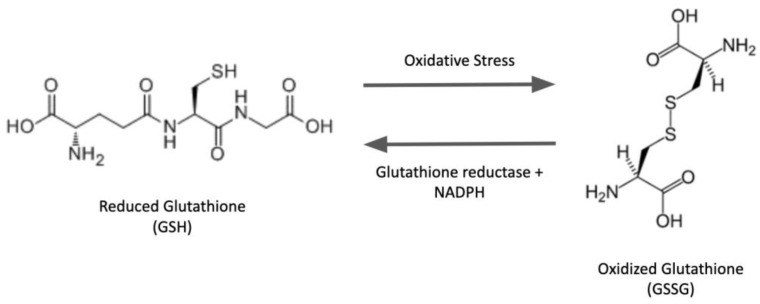
Structures of GSH and GSSG.

**Figure 2 antioxidants-12-01553-f002:**
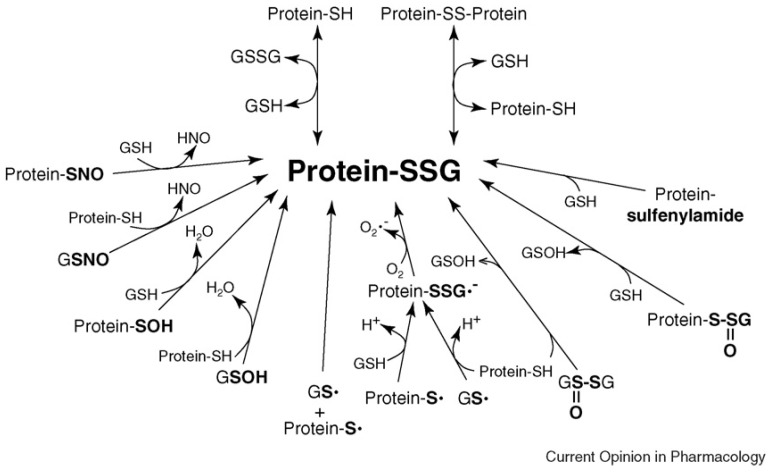
Likely mechanisms of protein glutathionylation within cells involve a reaction of the thiol moiety of protein-SH or GSH with a corresponding oxidized derivative; e.g., sulfenic acid (–SOH), thiyl radical (–S^•^), S-nitrosyl (–SNO), thiosulfinate (–S(O)SR), or sulfenyl-amide (cyclic-S–N–CO–). Oxidized sulfhydryl derivatives are proposed to form during normal redox signaling, pathological oxidative stress, and/or treatment with xenobiotic/pharmacological agents. Exposure of proteins and cultured cells to such reactive sulfhydryl intermediates is shown to lead to protein-SSG formation in vitro. Determining the importance of these glutathionylation mechanisms in vivo requires evaluation of the chemistry and biochemistry, including thermodynamic and kinetic competence, particularly under conditions that mimic the intracellular milieu. For additional details, see [24]. (Reproduced with permission [24]).

**Figure 3 antioxidants-12-01553-f003:**
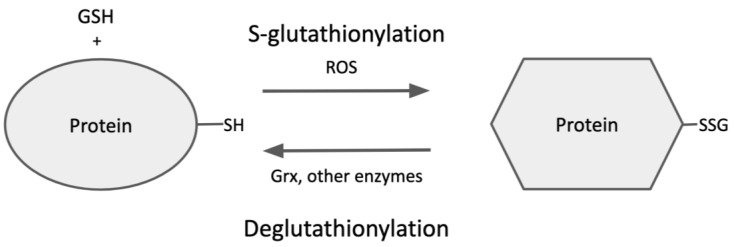
Protein S-glutathionylation and Deglutathionylation.

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
