# Peer review of "Glutathione and Glutaredoxin—Key Players in Cellular Redox Homeostasis and Signaling"

_antioxidants, 2023, doi:10.3390/antiox12081553_

Round 1

Reviewer 1 Report

The review of Chai and Mieyal is about Glutathione and Glutaredoxin. It is clearly written and summarized well this subject. However it is poorly illustrated. I have only minor comments.

In Text references. They have to be checked and written correctly (example L 78. 7,8,9,10,11,12,13,14 should be replaced by 7-14)

Abbreviations. Several abbreviations (Grx, ROS for examples) are introduced more than one time in the text. They have to be checked

References. DOI link have to be homogenized.

Overview. A figure with the molecules of reduced and oxidized glutathione would be helpful

L.91 and after. This part is not about glutaredoxin enzymes but more about protein glutathionylation

Scheme 1 should be centered        

L192 and after. A figure should be helpful

Part “deglutathionylation pf protein-SSG. A figure should be helpful

L299 and after. This is a separate part from the previous one. A title for this part have to be added.

Reviewer 2 Report

This is an excellent review article, which summarises the recent and previous finding on the glutathione/enzyme systems and their contribution to cell regulation and functions . The work is interesting and carefully written, however there are one point which can be addressed by the authors. 
The human GSTP1 enzyme should be written with the correct terminology, avoiding the Greek characters (old terminology). In addition, I would expect more information on the role of GSTs on the regulation and catalysis of glutathionylation. 
